# Utilization of Novel *Perilla* SSR Markers to Assess the Genetic Diversity of Native *Perilla* Germplasm Accessions Collected from South Korea

**DOI:** 10.3390/plants11212974

**Published:** 2022-11-03

**Authors:** Zhen Yu Fu, Kyu Jin Sa, Hyeon Park, So Jung Jang, Yeon Joon Kim, Ju Kyong Lee

**Affiliations:** 1Department of Applied Plant Sciences, College of Agriculture and Life Sciences, Kangwon National University, Chuncheon 24341, Korea; 2Interdisciplinary Program in Smart Agriculture, Kangwon National University, Chuncheon 24341, Korea

**Keywords:** *Perilla* crop, cultivated and weedy types, transcriptome RNA-seq, SSR marker, genetic diversity, phylogenetic relationship, population structure

## Abstract

The *Perilla* crop is highly regarded in South Korea, both as a health food and traditional food. However, there is still a lack of *Perilla* SSR primer sets (PSPSs) for studying genetic variation among accessions of cultivated and weedy types of *Perilla* crop (CWTPC) from South Korea. In this study, 30 PSPSs were newly developed based on transcriptome contigs in *P. frutescens*, and 17 of these PSPSs were used to study the genetic diversity, phylogenetic relationships and structure population among 90 accessions of the CWTPC collected from South Korea. A total of 100 alleles were detected from selected 17 PSPSs, with an average of 5.9 alleles per locus. The gene diversity (GD) ranged from 0.164 to 0.831, with an average of 0.549. The average GD values from the cultivated var. *frutescens*, weedy var. *frutescens*, and weedy var. *crispa*, were 0.331, 0.588, and 0.389 respectively. In addition, most variance shown by *Perilla* SSR markers was within a population (73%). An analysis of the population structure and phylogenetic relationships showed that the genetic relationship among accessions of the weedy var. *frutescens* and weedy var. *crispa* is closer than that for the accessions of the cultivated var. *frutescens*. Based on association analysis between 17 PSPSs and three seed traits in 90 *Perilla* accessions, we detected 11 PSPSs that together were associated with the seed size and seed hardness traits. Therefore, the newly developed PSPSs will be useful for analyzing genetic variation among accessions of the CWTPC, association mapping, and selection of important morphological traits in *Perilla* crop breeding programs.

## 1. Introduction

*Perilla frutescens* (L.) Britt. is an annual self-pollinating crop of the Labiatae family. The *Perilla* crop has been widely cultivated for a long time, mainly in East Asia, and is also widely distributed in Southeast Asia. In East Asia, the *Perilla* crop is divided into two varieties (or cultivated types) according to its morphological characteristics and availability as follows: cultivated var. *frutescens*, which is used as an oil, condiment, and vegetable (for wraps and salads) crop; and cultivated var. *crispa*, which is used as a Chinese herbal medicine and vegetable (for pickles and salads) crop [1,2,3,4,5]. Today, these two varieties (var. *frutescens*, var. *crispa*) of the *Perilla* crop are extensively cultivated and are used mainly in South Korea and Japan. In particular, var. *frutescens* is extensively cultivated and used mainly in South Korea, while var. *crispa* is extensively cultivated and used mainly in Japan [1,2,3,5,6,7]. However, the origin of the *Perilla* crop is believed to be China, because China has a long cultivation history, high genetic diversity, and many resources distributed in the country [1,2,3,4,8,9,10]. Although the wild species of the *Perilla* crop in East Asia have not yet been definitively identified, many weedy plants of the *Perilla* crop are widely distributed in China and South Korea [1,2,3,4,6,11]. In previous studies, Lee and Ohnishi (2001, 2003) [1,2], Lee et al. (2002) [6], Nitta and Ohnishi (1999) [12], and Nitta et al. (2003, 2005) [3,4] reported that the two cultivated types of *Perilla* crop have their weedy types, which grow naturally and are usually found in habitats such as wastelands, roadsides, stream or river sides, and the land around farmers’ houses and fields.

In East Asia, many researchers have conducted studies to distinguish between the accessions of cultivated and weedy types of the *Perilla* crop (CWTPC). The two cultivated types (var. *frutescens* and var. *crispa*) of the *Perilla* crop can be distinguished based on seed size, color of flower, color of leaf and stem, plant fragrance, and other morphological characteristics such as seed dormancy, seed hardness, pubescence degree, plant height, and leaf shape [1]. However, the two weedy types (var. *frutescens* and var. *crispa*) of the *Perilla* crop have not been clearly distinguished because of the presence of intermediate types [1]. The accessions of the CWTPC have the same number of chromosomes, 2n = 40 [13]. Additionally, cross experiments have been carried out by securing mating between accessions of the CWTPC using artificial pollination [14,15,16,17,18,19]. Therefore, it is important to understand the genetic variation and phylogenetic relationships between accessions of the CWTPC, and this will provide a rich source of information for making full use of germplasm resources in *Perilla* crop breeding programs.

In the *Perilla* crop, molecular marker technology has recently provided more comprehensive information on the genetic variation and phylogenetic relationships between and within accessions of the CWTPC. Molecular markers such as randomly amplified polymorphic DNA (RAPD) [3,12], amplified fragment length polymorphism (AFLP) [2,6], and simple sequence repeat (SSR) or microsatellite [5,7,19,20,21,22] have been successfully applied to accessions of the CWTPC, providing rich genetic diversity and genetic relationship information for genetic research of the *Perilla* crop. Among them, SSR markers are considered to be the preferred choices of markers for genetic studies because of their codominance, polymorphism, high repeatability, and rich genetic information [20,23]. In previous studies, developed *Perilla* SSR primers [19,21,24,25,26] have been used successfully to analyze the genetic variation and phylogenetic relationships as well as population structure between accessions of the CWTPC [5,7,11,18,19,21,22,27,28,29]. However, it is difficult to classify accessions of the CWTPC because the number of *Perilla* SSR primer sets (PSPSs) developed for analyzing the genetic variation and phylogenetic relationships of the CWTPC is still insufficient.

Recently, RNA-sequencing has presented a powerful tool for discovery of new genes, and the development of eSSR primers or microsatellite sites for non-model crops [30,31]. This RNA-seq transcriptome approach also facilitates rapid mining of SSR primers in non-model crops such as the *Perilla* species [21,26,32]. In the *Perilla* crop, Tong et al. (2015) [33] were the first to sequence and assembled one cultivated var. *frutescens* (PF98095) of the *Perilla* crop using transcriptome sequencing, conducted via RNA sequencing. In addition, Sa et al. (2018, 2019) [21,26] detected a total of 15,991 *Perilla* SSR loci, which were classified as dinucleotide SSRs, trinucleotide SSRs, and tetranucleotide SSRs based on the number of repeating units. In addition, using this information, novel PSPSs were developed in the *Perilla* crop [19,21,26].

Therefore, in this study, we developed novel PSPSs in the *Perilla* crop and performed genetic diversity, phylogenetic relationship and association mapping analysis on the CWTPC collected in South Korea, using the newly developed PSPSs. The current research results are expected to provide useful information for future genetic research, such as understanding the differentiation process of the CWTPC and genetic and breeding studies of *Perilla* germplasm accessions.

## 2. Results

### 2.1. Polymorphic Test for New Developed PSPSs

In the preliminary experiment, we surveyed the 200 newly developed PSPSs using four accessions of the *Perilla* species (see Appendix A). Among the newly designed 200 PSPSs, 30 PSPSs showed distinct amplification patterns and polymorphisms among the four *Perilla* accessions. However, the remaining PSPSs showed monomorphic bands (54), ambiguous banding patterns (31), and poor or no amplification banding patterns (85) (data not shown). Among the 30 PSPSs, 17 PSPSs presented clear band patterns that differentiate the *Perilla* species. Therefore, in our study, to understand the genetic variation among accessions of the CWTPC collected from South Korea using the newly developed PSPSs, we performed genetic diversity and phylogenetic relationship analysis among the 90 accessions of the CWTPC (30 cultivated var. *frutescens*, 30 weedy var. *frutescens*, 30 weedy var. *crispa*). In doing so, we made use of 17 PSPSs from among the newly developed 30 PSPSs (Table 1, Appendix A).

The 17 new PSPSs showed superior amplification patterns and polymorphisms among the 90 accessions of the CWTPC (Table 1, Appendix A). The 17 new PSPSs were used to determine polymorphisms in terms of the number of alleles, gene diversity (GD), major allele frequency (MAF), and polymorphic information content (PIC) among the 90 accessions of the CWTPC, which included cultivated var. *frutescens*, along with two weedy types of var. *frutescens* and var. *crispa* from South Korea (Table 1). A total of 100 alleles were detected from the 90 accessions of the CWTPC, which gave an average of 5.9 alleles per locus, ranging from 130 to 315 bp. The number of alleles per locus varied widely from 2 (KNUPF135) to 14 (KNUPF134). The MAF per locus ranged from 0.267 (KNUPF148) to 0.911 (KNUPF142), with an average of 0.587. GD values ranged from 0.164 (KNUPF 142) to 0.831 (KNUPF 134), with an average of 0.549. The average PIC was 0.507 and ranged from 0.153 (KNUPF142) to 0.817 (KNUPF134) (Table 1). Among the 100 alleles detected in the 90 accessions of the CWTPC, private alleles accounted for 23%, with rare alleles totaling 45%. Additionally, intermediate alleles accounted for 50% of the total alleles, while abundant alleles constituted 5% (Appendix A).

In this study, to compare the genetic variation of three groups of the CWTPC collected from South Korea, the 17 new PSPSs were used to determine genetic diversity in terms of the number of alleles, GD, MAF, and PIC among the 90 accessions of three groups of the CWTPC (Table 2). In a separate analysis of the three cultivated and weedy groups (30 cultivated var. *frutescens*, 30 weedy var. *frutescens*, 30 weedy var. *crispa*) of the 90 accessions of the CWTPC with the 17 new PSPSs, the average number of alleles was 3.3, 4.9, and 3.1 from the cultivated var. *frutescens*, weedy var. *frutescens*, and weedy var. *crispa*, respectively. The average MAF values were 0.749, 0.545, and 0.722 from the cultivated var. *frutescens*, weedy var. *frutescens*, and weedy var. *crispa*, respectively. The average GD values were 0.331 and 0.588 in the cultivated and weedy types of var. *frutescens*, and 0.389 in the weedy var. *crispa*. The average PIC values were 0.305 and 0.536 in the cultivated and weedy types of var. *frutescens* and 0.346 in the weedy var. *crispa* (Table 2).

### 2.2. Population Structure and Phylogenetic Relationship among Accessions of the CWTPC Collected from South Korea

To further understand the phylogenetic relationships of the CWTPC collected from South Korea, we analyzed the population structure and phylogenetic relationships among accessions of the CWTPC using newly developed PSPSs. In the population structure of the 90 accessions of the CWTPC, each accession was divided into its corresponding subgroups by the model-based approach in the STRUCTURE software. The highest value of *ΔK* for all 90 accessions of the CWTPC was for *K* = 2 (Figure 1a). At *K* = 2 and *K* = 3, all accessions of the CWTPC were divided into two and three principal groups, respectively (Figure 1b,c). Based on the membership probability threshold of 0.8, the 90 accessions of the CWTPC were divided into Group I, Group II, and an Admixed Group at *K* = 2 as well as Group I, Group II, Group III, and an Admixed Group at *K* = 3. In the results, at *K*= 2, Group I contained 24 accessions of weedy var. *frutescens* and 30 accessions of weedy var. *crispa.* Group II contained 29 accessions of cultivated var. *frutescens* and five accessions of weedy var. *frutescens*, and the Admixed Group contained one accession of cultivated var. *frutescens* and one accession of weedy var. *frutescens* (Figure 1b). At *K* = 3, Group I consisted of 24 accessions of weedy var. *crispa* and one accession of weedy var. *frutescens*. Group II consisted of 20 accessions of weedy var. *frutescens* and four accessions of weedy var. *crispa*. Group III consisted of 29 accessions of cultivated var. *frutescens* and five accessions of weedy var. *frutescens*. The Admixed Group consisted of one accession of cultivated var. *frutescens*, four accessions of weedy var. frutescens, and two accessions of weedy var. *crispa* (Figure 1c).

On the other hand, in the phylogenetic tree created using the dendrogram with an unweighted pair group method with arithmetic mean (UPGMA), the 90 accessions of the CWTPC were clustered into six major groups with 46.6% genetic similarity (Figure 2). Group I consisted of 30 accessions of cultivated var. *frutescens* and six accessions of weedy var. *frutescens*. Group II consisted of five accessions of weedy var. *frutescens* and 27 accessions of weedy var. *crispa*. Group Ⅲ consisted of eight accessions of weedy var. *frutescens* and one accession of weedy var. *crispa*. Group IV and Group V consisted of eight accessions of weedy var. *frutescens* and two accessions of weedy var. *frutescens*, respectively. Group VI consisted of one accession of weedy var. *frutescens* and two accessions of weedy var. *crispa*. Among all 90 accessions of the CWTPC, the accessions of the two weedy types of var. *frutescens* and var. *crispa* had the closest genetic relationship (Figure 2).

### 2.3. Seed Characteristics and Association Mapping Analysis of 90 Accessions of the CWTPC Using Novel PSPSs

The 90 accessions of the CWTPC used in this study showed significant differences in three seed characteristics, namely seed size (SS), seed hardness (SH), and seed coat color (SCC). As shown in Appendix A, for SS and SH, all accessions of cultivated var. *frutescens* showed a large size of 2 mm or more and soft seeds, whereas all accessions of the two weedy types of var. *frutescens* and var. *crispa* showed a small size of less than 2 mm and hard seeds. For SCC, the 30 accessions of cultivated var. *frutescens* showed white, gray, brown, and dark brown. Among them, three accessions were white, two accessions were gray, 13 accessions were brown, and 12 accessions were dark brown. Meanwhile, the 30 accessions of weedy var. *frutescens* showed brown and dark brown, of which 10 accessions were brown, and the remaining 20 accessions were dark brown. The 30 accessions of weedy var. *crispa* showed brown and dark brown, of which 7 were brown, 22 accessions were dark brown, and the remaining one accession had mixed brown and dark brown seeds (Appendix A).

In addition, to select the PSPSs associated with seed characteristics in the 90 accessions of the CWTPC, the 17 new PSPSs and three seed characteristics (SS, SH and SCC), were used to confirm significant marker–trait associations (SMTAs) using TASSEL 3.0 software. From the results, we detected 11 PSPSs (KNUPF132, KNUPF133, KNUPF136, KNUPF137, KNUPF138, KNUPF140, KNUPF141, KNUPF142, KNUPF145, KNUPF146, and KNUPF148) that were together associated with the SS and SH traits using GLM at a significance level of *p* ≤ 0.01 (Table 3). Additionally, four PSPSs (KNUPF134, KNUPF135, KNUPF145, and KNUPF146) were related to the SCC trait using GLM at a significance level of *p* ≤ 0.05, but not at a significance level of *p* ≤ 0.01 (Table 3).

## 3. Discussion

### 3.1. Development of Novel SSR Markers in Perilla Crop and Their Use for Genetic Variation Analysis

Understanding the genetic variation of plant genetic resources in crop species is necessary for the conservation of biodiversity and the development of new cultivars of crop species. In recent years, PSPSs have been widely used to identify genetic diversity and phylogenetic relationships, as well as for analysis of population structure and association mapping among accessions of the CWTPC [7,11,18,19,21,22,27,29]. However, as mentioned in the Introduction, the number of SSR markers developed in the *Perilla* crop is still insufficient. That is, the classification of the CWTPC is not clear because of a lack of PSPSs for analyzing the genetic variation of the *Perilla* crop and its related weedy types. Therefore, the new PSPSs developed in the current study provide information for further determination of the genetic variation among accessions of the CWTPC, and such information may be useful in understanding the genetic differentiation and phylogenetic relationships of accessions of the CWTPC collected from South Korea. In particular, in recent years the cultivated var. *frutescens* of the *Perilla* crop has become one of the most important economic crops in South Korea because of an increase in oil consumption, along with its use as a leaf vegetable. Therefore, genetic diversity analysis using the newly developed PSPSs on the *Perilla* germplasm resources collected in South Korea will provide good information for the selection of useful germplasm resources, the conservation of germplasm resources of the CWTPC, the differentiation study of the CWTPC, and the development of the *Perilla* cultivars (or varieties) in future studies.

As already mentioned in the Introduction and Materials and Methods sections, in the case of the *Perilla* crop, Sa et al. (2018) [21] reported that most SSRs were dinucleotide SSRs (9910), which occupied 62.0% of all SSR loci based on the number of repeating units among a total of 15,991 SSR loci [33]. Therefore, the dinucleotide repetitions were more abundant than other repeat units in the CWTPC. This information is very important for developing a large number of effective eSSR primers or microsatellite loci in the CWTPC because the SSR primers associated with short microsatellite motif lengths are more variable than those associated with long microsatellite motif lengths [34]. Therefore, in this study, we developed 30 PSPSs that were dinucleotide SSRs and used 17 of these PSPSs to analyze the genetic variation and phylogenetic relationships between and within the accessions of the CWTPC collected from South Korea. From our experimental results, a total of 100 alleles were obtained in 90 accessions of the CWTPC, with an average of 5.9 alleles per locus (Table 1). This value is lower than the number of alleles per SSR locus that was detected in some previous studies on the *Perilla* crop, such as the value of 9.2 detected by Sa et al. (2013) [7], the 9.8 detected by Sa et al. (2015) [35], and the 7.9 detected by Sa et al. (2021) [5]; but it is slightly higher than the 4.5 detected by Ma et al. (2017) [36], 3.3 detected by Kim et al. (2021) [19], and 5.8 detected by Park et al. (2021) [37]. The average GD value obtained in this study was similar to that in previous studies at 0.549 (Table 1), which is lower than the results detected by Sa et al. (2013, 2015, 2021) [5,7,35], Park et al. (2021) [37], and Kim et al. (2021) [19] and slightly higher than the results detected by Ma et al. (2017) [36]. Generally, a GD value > 0.5 shows that the screened SSR primer loci had higher genetic resolution and that the studied accessions of the CWTPC also had higher genetic diversity [38,39]. Furthermore, the genetic diversity between the accessions of three cultivated and weedy groups (30 accessions of cultivated var. *frutescens*, 30 accessions of weedy var. *frutescens*, 30 accessions of weedy var. *crispa*) of the CWTPC was analyzed using the 17 PSPSs. The number of these accessions of the CWTPC is the most used for diversity analysis research in South Korea so far. Therefore, the results of this study obtained using the new PSPSs are expected to provide useful information for understanding the genetic diversity and phylogenetic relationship of the accessions of the CWTPC in South Korea. As a result, the average GD values of the accessions of cultivated and weedy types of var. *frutescens* and weedy var. *crispa* from South Korea were 0.331, 0.588, and 0.389, respectively (Table 2). From the results, accessions from South Korea of weedy var. *frutescens* showed the highest genetic variation compared with the accessions of cultivated var. *frutescens* and weedy var. *crispa.* In addition, accessions of the two weedy types of var. *frutescens* and var. *crispa* showed much higher genetic variation than the accessions of cultivated var. *frutescens*. These results are consistent with previous reports of analysis results by Lee and Ohnishi (2003) [2] and Lee et al. (2002) [6] using AFLP markers, as well as those of Sa et al. (2013) [7] using SSR markers. The reason for these results may be that many allelic genes were lost by natural or human selection during the evolution from wild to cultivated species within and between the CWTPC.

In crop species, seed characteristics are one of the useful traits that distinguish cultivated type from wild or weedy types. For example, SS, SH, seed dispersal, and seed dormancy (SD) differ dramatically between cultivated grain crops and their wild ancestors [22,40,41,42,43]. In the case of the *Perilla* crop, seed characteristics, especially SS, SH, SCC, and SD showed great differences between the CWTPC [1,21,22]. In our study, the 90 accessions of the CWTPC used in this study showed significant differences in the three seed characteristics of SS, SH, and SCC (Appendix A). According to our results, the cultivated var. *frutescens* showed a large seed size of 2 mm or more, whereas the two weedy types of var. *frutescens* and var. *crispa* showed a small seed size of less than 2 mm. For SH, most accessions of cultivated var. *frutescens* showed soft seed characteristics, whereas all accessions of the two weedy types of var. *frutescens* and var. *crispa* showed hard seed characteristics. For SCC, cultivated var. *frutescens* showed various seed coat colors such as white, gray, brown, and dark brown, whereas the two weedy types of var. *frutescens* and var. *crispa* showed only brown and dark brown seed coat colors. However, there was no significant difference between the two weedy types of var. *frutescens* and var. *crispa* of the *Perilla* crop (Appendix A). In addition, most accessions of the two weedy types of var. *frutescens* and var. *crispa* had seed dormancy, but most accessions of the cultivated var. *frutescens* did not have seed dormancy and showed a high germination rate [21,22]. In particular, seed dormancy is thought to be a strongly selected trait during domestication [40]. Therefore, among the CWTPC, cultivated var. *frutescens* was considered to be the most evolved ecotype through human cultivation. As previously reported by Lee and Ohnishi (2003) [2], South Korea is considered as an important genetic diversity center for the CWTPC in East Asia. In the case of soybean and foxtail millet crops, although the origin of these crops is China, many wild-type or weed-type accessions are widely distributed in South Korea [44,45]. Therefore, although no wild species of the *Perilla* crop have been found in East Asia, these accessions of the two weedy types of the *Perilla* crop are thought to have an important place in the origin of the two cultivated types of the *Perilla* crop. Therefore, the novel PSPSs developed in this study were expected to provide useful information for the evaluation of genetic variation in accessions of the CWTPC collected from South Korea.

### 3.2. Analysis of Genetic Diversity, Phylogenetic Relationships and Association Mapping Analysis of the CWTPC Collected in South Korea

In order to understand further the phylogenetic relationship between and within accessions of the CWTPC collected from South Korea, the population structure and phylogenetic relationships among accessions of the CWTPC were analyzed. We adopted the ad hoc measure *ΔK* proposed by Evanno et al. (2005) [46]. Based on these criteria, we measured *ΔK* = 2 (Figure 1a), and because there are three groups of *Perilla* accessions, we analyzed the population structure based on *K* = 2 and *K* = 3. At *K* = 2, all accessions of the CWTPC were divided into two groups as follows: most accessions of the weedy var. *frutescens* and all accessions of the weedy var. *crispa* formed one group, while almost all accessions of the cultivated var. *frutescens* and five accessions of the weedy var. *frutescens* formed one group (Figure 1b). This result shows that the genetic relationship between accessions of the two weedy types of var. *frutescens* and var. *crispa* is closer than that for the accessions of cultivated var. *frutescens*. A phylogenetic relationship analysis (Figure 2) also showed the same results. At *K* = 3, all accessions of the CWTPC were divided into Group I, Group II, Group III, and an Admixed Group. Except for ambiguous accessions in each group, Group I consisted mostly of accessions of weedy var. *crispa*, Group II consisted mostly of accessions of weedy var. *frutescens*, Group III consisted mostly of accessions of cultivated var. *frutescens*, and the Admixed Group included seven accessions of the CWTPC (Figure 1c). Therefore, with the exception of some ambiguous accessions, most accessions of the CWTPC were clearly distinguished by the population structure analysis as the three groups.

On the other hand, in the analysis of the UPGMA dendrogram for accessions of the CWTPC (Figure 2), the clustering patterns did not show a clear classification between the cultivated and weedy types of var. *frutescens* and the weedy var. *crispa*. Although most accessions of the two weedy types of var. *frutescens* and var. *crispa* did not form different groups, all accessions of cultivated var. *frutescens* were clearly different from the two weedy types of var. *frutescens* and var. *crispa*. These results indicate that, as with morphological characteristics such as seed traits (see Appendix A), SSR markers can be used to distinguish between accessions of the CWTPC. Additionally, these results are consistent with previous analysis results of AFLP markers by Lee et al. (2002) [6] and Lee and Ohnishi (2003) [2]. In addition, in Group I of the UPGMA dendrogram (Figure 2), the six accessions of weedy var. *frutescens* were very closely related to all the accessions of cultivated var. *frutescens*. According to a previous report by Lee et al. (2002) [6] and Lee and Ohnishi (2003) [2], these exceptional accessions of the weedy var. *frutescens* could be an escaped form from cultivation, or otherwise likely derived from hybrids between accessions of the CWTPC. Although there is still no accurate report on the natural hybridization rate in the *Perilla* crop and its weedy varieties, recently Lim et al. (2019, 2021) [17,18] and Kim et al. (2021) [19] reported artificial hybridization between cultivated var. *frutescens* and weedy var. *crispa*.

In addition, most accessions of the two weedy types of var. *frutescens* and var. *crispa* were well distinguished from each other in the UPGMA dendrogram, but some accessions were poorly differentiated (Figure 2). Lee and Ohnishi (2003) [2] reported the reason for such results in a previous study revealed by AFLP markers as follows: the two weedy types of var. *frutescens* and var. *crispa* of the *Perilla* crop originated from a common ancestor. Weedy forms have been reported in many crops such as rice, soybean, barley, oat, sorghum, and foxtail millet [40,44,45,47]. In many crop species, weedy types have generally been regarded as either wild ancestors of crops or escape forms of cultivated crops. Research by Lee et al. (2002) [6] and Lee and Ohnishi (2003) [2] using AFLP markers and Lee and Kim (2007) [48], Sa et al. (2013, 2018) [7,21], and Ma et al. (2019) [11] using SSR markers suggests that the weedy-type accessions of the *Perilla* crop in East Asia are the key taxon in understanding the origin of the *Perilla* crop. In our analysis, some accessions of the two weedy types of var. *frutescens* and var. *crispa* were closely related and difficult to distinguish from each other. In addition, as shown in Table 4, most variance (73%) shown by the *Perilla* SSR markers was within the population and the variance observed among the populations was relatively small (27%). The present study revealed that genetic variation occurred mainly within the population rather than among the three populations, as shown by the population structure and UPGMA dendrogram analyses (Figure 1 and Figure 2). This indicates that variations in the CWTPC were mainly because of the influences of variation caused by gene flow or gene exchange between and within the population of the CWTPC. Finally, in our study, in order to select the PSPSs associated with seed characteristics among 90 accessions of the CWTPC, we analyzed marker trait associations between 17 PSPSs and three seed characteristics (SS, SH, and SCC) in 90 accessions of the CWTPC using TASSEL software (Table 3). From the results, we identified 11, 11, and 4 SSR markers associated with SS, SH, and SCC, respectively. Among these SSR markers related to seed characteristics, 11 PSPSs (KNUPF132, KNUPF133, KNUPF136, KNUPF137, KNUPF138, KNUPF140, KNUPF141, KNUPF142, KNUPF145, KNUPF146, and KNUPF148) were together associated with the SS and SH traits. In particular, three PSPSs (KNUPF135, KNUPF145, and KNUPF146) were together associated with the SS, SH, and SCC traits. Although, these SSR markers are thought to be useful molecular markers for distinguishing seed characteristics in the CWTPC, precise analysis using a genetic mapping population or more research materials are required.

Today the var. *frutescens* is extensively cultivated and used in South Korea (see Introduction). In particular, many accessions of the CWTPC were found throughout South Korea [1,3,4,5,6,11,28]. This current study, for more accurate genetic diversity and phylogenetic analysis of the CWTPC in South Korea, was performed using more *Perilla* germplasm accessions and novel PSPSs compared with previous reports by Lee et al. (2002) [6], Lee and Ohnishi (2003) [2], Sa et al. (2013, 2018) [7,21], Lee and Kim (2007) [48], and Ma et al. (2019) [11]. Therefore, the newly developed PSPSs will be useful for molecular breeding studies such as analysis of genetic diversity between accessions of the CWTPC, association and QTL mapping, and selection of important morphological traits in the *Perilla* crop.

## 4. Materials and methods

### 4.1. Plant Materials and DNA Extraction

In the preliminary experiment of our study, four accessions of the *Perilla* crop (one accession of cultivated var. *frutescens*, one accession of weedy var. *frutescens*, one accession of cultivated var. *crispa*, and one accession of weedy var. *crispa*) were used for the selection of 200 newly developed PSPSs (Appendix A). In addition, for genetic diversity and phylogenetic relationship analysis of South Korean *Perilla* germplasm accessions using the selected PSPSs, a total of 90 accessions of the CWTPC (30 cultivated var. *frutescens*, 30 weedy var. *frutescens*, 30 weedy var. *crispa*) were selected from the published germplasm on the website of the Korean RDA-Genebank of the Republic of Korea (https://genebank.rda.go.kr/, accessed on 1 March 2021). The 90 accessions of the CWTPC used in this study were selected to represent, as much as possible, all regions of South Korea, and the collection information is shown in Appendix A. In addition, the seed characteristics, especially SS, SH, and SCC of the 90 accessions of CWTPC used in this study were investigated according to the previous research method of Lee and Ohnishi (2001) [1]. Information on the accession numbers of the CWTPC and their seed characteristics are shown in Appendix A. The total DNA of each *Perilla* accession was extracted from the young leaves at the seedling stage based on a modified Doyle’s CTAB method [49].

### 4.2. Development of PSPSs and SSR Amplification

To develop PSPSs based on the previous published results of Tong et al. (2015) [33], we performed de novo assembly of the accession of cultivated var. *frutescens* (PF98095) RNA-seq data using Trinity software (http://TrinityRNASeq.sourceforge.net) (accessed on 1 January 2021). As previously reported by Sa et al. (2018, 2019) [21,26] and Kim et al. (2021) [19], we filtered the raw reads from Next Generation Sequencing (NGS) with a Phred quality score of at least 20 and read length of at least 50 bp of HiSeq 2000 data before assembly. A Perl script MISA tool (http://pgrc.ipk-gatersleben.de/misa) (accessed on 1 January 2021) was used to search for new *Perilla* SSR or microsatellite sites in the assembled transcriptome sequences of the accession of cultivated var. *fruteseces* (PF98095) of the *Perilla* crop. In accordance with the *Perilla* SSR flanking sequences of PF98095, new PSPSs were designed using PRIMER 3 software. In previous studies, Sa et al. (2018, 2019) [21,26] detected a total of 15,991 *Perilla* SSR loci (100%), which were classified as dinucleotide SSRs (9910, 62%), trinucleotide SSRs (5652, 35.3%), and tetranucleotide SSRs (429, 2.7%) based on the number of repeating units. Therefore, in this study, we designed 200 PSPSs from the 9910 dinucleotide SSRs based on the number of di-repeat units. The SSR amplification method for accessions of the CWTPC has been described in a previous study by Sa et al. (2018, 2019) [21,26]. After polymerase chain reaction (PCR) amplification using PSPSs, DNA electrophoresis analysis was performed according to the method described in a previous study by Sa et al. (2018, 2019) [21,26].

### 4.3. Data Analysis

DNA fragments amplified by PSPSs were scored as present (1) or absent (0). The number of alleles, allele frequency, GD, MAF, and PIC for the 17 PSPSs was calculated with the PowerMarker 3.25 software program [50]. Genetic similarities (GS) were calculated for each pair of accessions using the Dice similarity index [51]. To analyze the phylogenetic relationship of the 90 accessions of the CWTPC, a similarity matrix was used to construct a dendrogram with an UPGMA by SAHN-Clustering from NTSYS-pc V2.1 software [52]. The population structure of the 90 accessions of the CWTPC was investigated by using STRUCTURE 2.2 software [53]. Each simulated value of k = 1–10 was run independently for five times, the burn-in period was 100,000 runs, and then 100,000 Markov Chain Monte Carlo (MCMC) replications were performed. The delta *K* statistic was based on the rate of change of the logarithmic probability of the data between *K* values [46], which was calculated from STRUCTURE HARVESTER (http://taylor0.biology.ucla.edu/structHarvester/, accessed on 1 December 2021) based on the STRUCTURE results. Association mapping analysis was performed for the 17 novel PSPSs and three seed characteristics (SS, SH, SCC) of the 90 accessions of the CWTPC to confirm SMTAs using TASSEL 3.0 software [54], which was used to evaluate marker–trait associations using a general linear model (Q GLM). The Q GLM method used in this study was performed according to the method described in a previous study by Ha et al. (2021) [22] and Park et al. (2021) [37].

## 5. Conclusions

The *Perilla* crop is highly regarded as a health food and traditional food in South Korea owing to the use of its seeds and leaves. However, PSPSs for studying genetic variation among accessions of the CWTPC from South Korea are still lacking. In this study, we developed 30 PSPSs (dinucleotide SSRs) using transcriptome sequencing by RNA-sequencing and used 17 of these PSPSs to analyze the genetic variation and phylogenetic relationships between and within 90 accessions of the CWTPC collected from South Korea. The number of these accessions of the CWTPC is the largest number used so far for diversity analysis research in South Korea. Therefore, the results of this study obtained using the new PSPSs are expected to provide a useful information for understanding the genetic diversity and phylogenetic relationship of the accessions of the CWTPC in South Korea. In our study, 100 alleles in total were detected from the 90 accessions of the CWTPC with an average of 5.9 alleles per locus, ranging from 130 to 315 bp. The average GD values of the accessions of cultivated and weedy types of var. *frutescens* and weedy var. crispa from South Korea were 0.331, 0.588, and 0.389, respectively. The results showed that the accessions of weedy var. *frutescens* had the highest genetic variation compared with the accessions of cultivated var. *frutescens* and weedy var. *crispa*. In the *Perilla* crop, seed characteristics, especially SS, SH, and SCC, showed great differences between the CWTPC. Of the 17 PSPSs used in the analysis, 11 PSPSs (KNUPF132, KNUPF133, KNUPF136, KNUPF137, KNUPF138, KNUPF140, KNUPF141, KNUPF142, KNUPF145, KNUPF146, and KNUPF148) were together associated with the SS and SH traits. According to analysis of the seed characteristics, population structure and phylogenetic relationship, although most accessions of the two weedy types of var. *frutescens* and var. *crispa* did not form different groups, all accessions of cultivated var. *frutescens* were clearly different from the two weedy types of var. *frutescens* and var. *crispa*. In this study, most variance shown by the *Perilla* SSR markers was within a population (73%) so that the genetic variation occurred mainly within the population of the CWTPC. Therefore, the newly developed PSPSs will be useful for analyzing genetic variation among accessions of the CWTPC, association and QTL mapping, and selection of important morphological traits in *Perilla* crop breeding programs.

## Figures and Tables

**Figure 1 plants-11-02974-f001:**
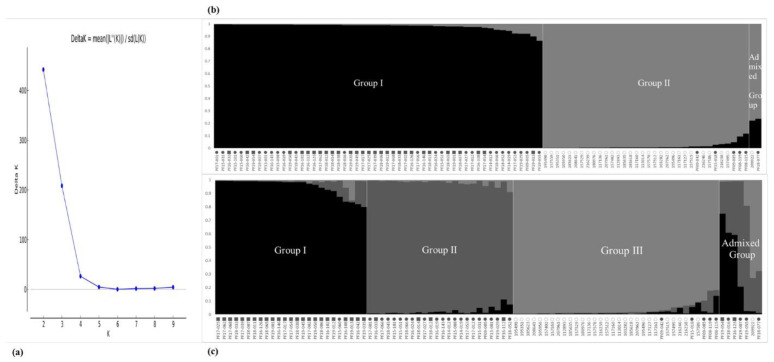
Magnitude of *∆K* as a function of *K*; the peak value of *∆K* was at *K* = 2 (**a**). Population structure of 90 *Perilla* accessions collected from South Korea based on new 17 *Perilla* SSRs for *K* = 2 (**b**) and *K* = 3 (**c**); ○: accessions of cultivated var. *frutescens*, ●: accessions of weedy var. *frutescens*, ■: accessions of weedy var. *crispa*.

**Figure 2 plants-11-02974-f002:**
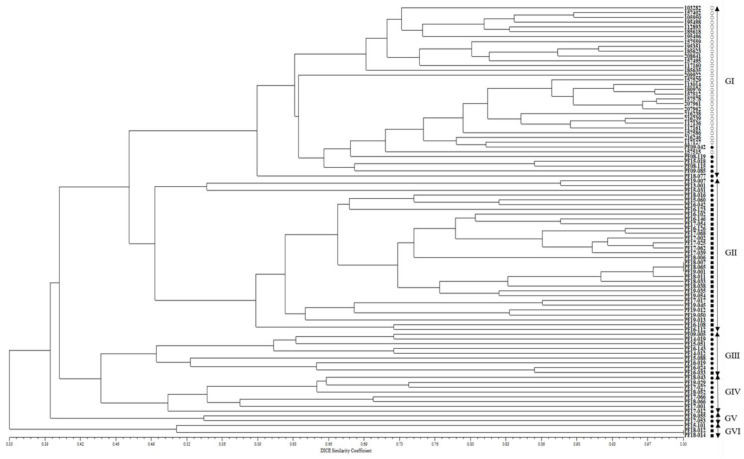
UPGMA dendrogram of 90 *Perilla* accessions collected from South Korea based on new 17 *Perilla* SSR markers. ○: accessions of cultivated var. *frutescens*, ●: accessions of weedy var. *frutescens*, ■: accessions of weedy var. *crispa*.

**Table 1 plants-11-02974-t001:** Characteristics of the 17 SSR loci including primer sequence, repeat motif, annealing temperature, allele size range, genetic diversity index among 90 *Perilla* accessions.

SSR Loci	Forward Primer (5’–3’)	Reverse Primer (3’–5’)	Repeat Motif	T_a_	Allele Size (bp)	Allele No	MAF	GD	PIC
KNUPF127	CTGAGCAGAATGGGATAAAATC	CATGAATCCAAACCTGAGAAAT	(AT)_8_	61	210–240	5	0.778	0.378	0.356
KNUPF128	TTTTCTGGAAAGAAAAACCAAA	GTCATTTTCCAAACCGTAAAAA	(AT)_8_	60	160–180	9	0.544	0.597	0.532
KNUPF129	AATACATGAACACTGTCACACCA	AGATCATGTTAGCAGGCAATTT	(AT)_8_	64	155–180	14	0.344	0.831	0.817
KNUPF130	TGAGAAATCTAACCCCAAACTT	CCTGTTTTTGATCTCTTACTTGC	(CA)_8_	62	200–220	2	0.778	0.346	0.286
KNUPF131	TGGATCAAACATTGTAACAGGA	ACCAACACCAAAACTACTGACC	(CA)_8_	63	190–220	3	0.778	0.370	0.340
KNUPF132	TTTGAGATAGCTCGGTTCAAAT	CTTCAGGAGCCACATATTCTTC	(AG)_16_	62	190–230	9	0.478	0.709	0.678
KNUPF133	TTAAAAGATTGCATGTCTGCAC	CCTTTTCCTGTGTTTTCTCAAG	(AG)_16_	62	220–260	3	0.456	0.592	0.505
KNUPF134	TATAATACACGAAGACGCCACA	TTTTGTCCTGTCAACTTCCTCT	(AG)_15_	64	130–150	6	0.367	0.749	0.710
KNUPF135	AATAGGTCGACTATGTTCGTGG	ATCAAATCTGCCAATCTCATTT	(CT)_12_	62	135–175	6	0.811	0.332	0.318
KNUPF136	TCAAGCAGAGATTGATTCAGTG	CAAAGAATAATCACCACACCAA	(AG)_12_	62	140–170	6	0.589	0.589	0.544
KNUPF137	AATCAAGGTGTGCAATCATACA	GGTGTTCACTAGAGTCTCGGTC	(CT)_11_	64	300–315	3	0.911	0.164	0.153
KNUPF138	CTGCGTGTGCTGATAAAACTC	TTCTGCTGCTGTATTCTGAGTG	(AG)_11_	64	150–180	5	0.556	0.607	0.554
KNUPF139	CCCTAAATCAAACTTGAATCCC	GGGTCGCTAGTAAAGAAGGTTT	(CT)_10_	62	170–190	6	0.567	0.598	0.545
KNUPF140	GGGTTCTTTCTTTCTCCCTTTA	AGCTAAGCTGGCTTCTCTATTTT	(CT)_10_	63	200–220	5	0.689	0.448	0.374
KNUPF141	ATCTTTCGCAATATGTTTCCTG	AAGTTCACAAAGTTGAACGCTT	(CT)_10_	63	180–190	4	0.656	0.515	0.466
KNUPF142	ATCTCGCATTCTTTTAGCTACG	TTTCTCGGAAAATCACTCTGTT	(CT)_10_	62	200–230	6	0.411	0.713	0.665
KNUPF143	GGATCTTCTGGGATTTCTTACC	GCCGTATGTCGTCCTTGAT	(CT)_10_	63	165–190	8	0.267	0.800	0.770
Average						5.9	0.587	0.549	0.507

Note: T_a_—Annealing Temperature, MAF—Major Allele Frequency, GD—Gene Diversity, PIC—Polymorphism Information Content.

**Table 2 plants-11-02974-t002:** Genetic variation obtained from each SSR locus among 90 accessions of *Perilla* crop and their weedy types collected from South Korea.

SSR Loci	Cultivated var. *frutescens* (n = 30)				Weedy var. *frutescens* (n = 30)				Weedy var. *crispa* (n = 30)			
	Allele No	MAF	GD	PIC	Allele No	MAF	GD	PIC	Allele No	MAF	GD	PIC
KNUPF127	2	0.933	0.124	0.117	4	0.833	0.296	0.282	4	0.567	0.562	0.486
KNUPF128	4	0.633	0.536	0.484	6	0.533	0.618	0.561	5	0.467	0.591	0.507
KNUPF129	8	0.300	0.796	0.766	10	0.267	0.851	0.835	4	0.733	0.429	0.393
KNUPF130	1	1.000	0.000	0.000	2	0.567	0.491	0.371	2	0.767	0.358	0.294
KNUPF131	1	1.000	0.000	0.000	3	0.733	0.418	0.370	3	0.600	0.540	0.466
KNUPF132	6	0.433	0.664	0.605	9	0.567	0.644	0.622	4	0.833	0.293	0.276
KNUPF133	2	0.967	0.064	0.062	3	0.400	0.640	0.563	2	0.900	0.180	0.164
KNUPF134	4	0.767	0.393	0.371	6	0.533	0.618	0.561	3	0.567	0.540	0.450
KNUPF135	1	1.000	0.000	0.000	5	0.567	0.609	0.561	3	0.867	0.238	0.221
KNUPF136	4	0.533	0.620	0.561	4	0.433	0.638	0.568	4	0.833	0.293	0.276
KNUPF137	2	0.933	0.124	0.117	3	0.800	0.331	0.294	1	1.000	0.000	0.000
KNUPF138	1	1.000	0.000	0.000	5	0.567	0.627	0.591	4	0.633	0.551	0.511
KNUPF139	3	0.900	0.184	0.175	5	0.600	0.589	0.551	3	0.633	0.531	0.475
KNUPF140	4	0.867	0.242	0.232	3	0.567	0.518	0.414	2	0.800	0.320	0.269
KNUPF141	3	0.600	0.540	0.466	3	0.533	0.598	0.526	2	0.833	0.278	0.239
KNUPF142	4	0.500	0.620	0.551	6	0.433	0.720	0.680	2	0.900	0.180	0.164
KNUPF143	6	0.367	0.724	0.679	6	0.333	0.791	0.762	5	0.333	0.733	0.686
Mean	3.3	0.749	0.331	0.305	4.9	0.545	0.588	0.536	3.1	0.722	0.389	0.346

Note: MAF—Major Allele Frequency, GD—Gene Diversity, PIC—Polymorphism Information Content.

**Table 3 plants-11-02974-t003:** Information on SMTA markers using GLM method for 90 *Perilla* accessions.

Trait	SSR Marker	GLM	SSR Marker	GLM
SS	KNUPF132	**	KNUPF141	**
	KNUPF133	**	KNUPF142	**
	KNUPF135	*	KNUPF143	*
	KNUPF136	**	KNUPF145	**
	KNUPF137	**	KNUPF146	**
	KNUPF138	**	KNUPF147	*
	KNUPF139	*	KNUPF148	**
	KNUPF140	**		
SH	KNUPF132	**	KNUPF141	**
	KNUPF133	**	KNUPF142	**
	KNUPF135	*	KNUPF143	*
	KNUPF136	**	KNUPF145	**
	KNUPF137	**	KNUPF146	**
	KNUPF138	**	KNUPF147	*
	KNUPF139	*	KNUPF148	**
	KNUPF140	**		
SCC	KNUPF134	*	KNUPF145	*
	KNUPF135	*	KNUPF146	*

* *p* ≤ 0.05, ** *p* ≤ 0.01.

**Table 4 plants-11-02974-t004:** Analysis of molecular variance (AMOVA) based on SSR marker among 90 accessions of cultivated and weedy types of the *Perilla* crop collected from South Korea.

Source of Variation	Degree of Freedom (df)	Sum of Square (SS)	Variance Component	Percentage of Total Variance	*p* Value
Among Pop	2	176.16	2.70	27	0.275
Within Pop	87	619.37	7.12	73	0.001
Total	89	795.52	9.82	100	

Level of significance based on 999 permutations; *p* value = probability of obtaining a more extreme component by chance alone, Pop population.

## Data Availability

All data generated or analyzed during this study are included in this published article and its Appendix A.

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
