# Peer review of "Utilization of Novel *Perilla* SSR Markers to Assess the Genetic Diversity of Native *Perilla* Germplasm Accessions Collected from South Korea"

_plants, 2022, doi:10.3390/plants11212974_

Round 1
Reviewer 1 Report
the paper could be published in its present form.
Author Response
The paper could be published in its present form.
-> Thank you for your comments.

Reviewer 2 Report
This is an interesting study about Genetic diversity and phylogenetic relationships among accessions of cultivated and weedy types of Perilla frutescens (L.) Britton collected from South Korea using newly developed SSR markers.
The study updates us with 30 newly developed SSR markers using transcriptome sequencing with RNA-sequencing, where 17 of these SSRs were used to study the genetic diversity, phylogenetic relationships and structure population among 90 accessions of the CWTPC collected from South Korea. The readability of the manuscript is fluent and understandable. There are some concerns which should be addressed before accepting for publication:
1. Additional English editing should be done; please check the use of prepositions (I am not a native English speaker but there might be some corrections needed for the final tuning of the manuscript).
2. Please explain the criteria used for the selection of 17 SSR marker set (out of 30) to validate them and evaluate the population genetics parameters since you already described the criteria for the selection of SSR marker from 200 to 30 (pilot experiment)…
3. There is no need to define GD and PIC formulas in the MS since the algorithms are known and already implemented in the software(s) used and citied for population genetics analysis.
4. Please cite the references correctly. Structure harvester has his own authors, so there is no need to cite with url address.
5. In this study I would suggest to calculate HWE for each SSR marker (out of 17) separately to specific weedy type since average PIC values are moderate/low. I would also suggest including the values of null alleles for each locus (17) calculated for the selected set of 90 accessions.
6. Please check the style and references according to Plants journal requirements. It seems data different fonts were used in the MS.
Author Response
Reviewer #2:
This is an interesting study about Genetic diversity and phylogenetic relationships among accessions of cultivated and weedy types of Perilla frutescens (L.) Britton collected from South Korea using newly developed SSR markers.
The study updates us with 30 newly developed SSR markers using transcriptome sequencing with RNA-sequencing, where 17 of these SSRs were used to study the genetic diversity, phylogenetic relationships and structure population among 90 accessions of the CWTPC collected from South Korea. The readability of the manuscript is fluent and understandable. There are some concerns which should be addressed before accepting for publication:
- Additional English editing should be done; please check the use of prepositions (I am not a native English speaker but there might be some corrections needed for the final tuning of the manuscript).
-> The revised manuscript was corrected in English by a native speaker.
- Please explain the criteria used for the selection of 17 SSR marker set (out of 30) to validate them and evaluate the population genetics parameters since you already described the criteria for the selection of SSR marker from 200 to 30 (pilot experiment)…
-> In the revised manuscript, we explained ‘why we chose 17 SSR marker sets from 30 SSR marker sets’ as follow: In addition, among the 30 PSPSs selected in the preliminary experiment, finally, we used 17 SSR primers for genetic diversity analysis among germplasm accessions of the CWTPC collected from South Korea, because they showed the most distinct amplification patterns and polymorphisms between the four accessions of the CWTPC. For reference, Supplementary Figure 1 showed the amplification pattern of the 200 SSR primers developed in this experiment. See L. 107-111, in new version manuscript.
- There is no need to define GD and PIC formulas in the MS since the algorithms are known and already implemented in the software(s) used and citied for population genetics analysis.
->We deleted ‘GD and PIC formulas’ in the MS. See L. 454-472, in new version manuscript.
- Please cite the references correctly. Structure harvester has his own authors, so there is no need to cite with url address.
-> We changed the references correctly. See L. 465-466, 638-639, in new version manuscript.
- In this study I would suggest to calculate HWE for each SSR marker (out of 17) separately to specific weedy type since average PIC values are moderate/low. I would also suggest including the values of null alleles for each locus (17) calculated for the selected set of 90 accessions.
-> We appreciate your review comments. As shown in the Supplement Table below, the number of null alleles at each locus was different for the 17 SSR primers. According to testing HWE for each locus among the three populations of 90 accessions of the CWTPC, all markers did not follow Hardy-Weinberg Equilibrium at p < 0.05.
The statistical analysis methods used in this study are already commonly used by all researchers. Therefore, we are sure that there are no special problems with the current analysis results. In particular, our research team has expertise in analyzing genetic diversity in plant populations using SSR markers in Perilla species, and has published many papers so far.
Supplement Table. Null alleles and HWE (P values) obtained from each SSR locus among the three populations of 90 accessions of the CWTPC collected from South Korea.
- Please check the style and references according to Plants journal requirements. It seems data different fonts were used in the MS.
-> We changed the references according to Plants journal requirements. See L. 529-641, in new version manuscript.
Thank you for your review comments.

Reviewer 3 Report
This manuscript reports new EST-SSR markers in Perilla frutescens and genotyping in a South Korean Perilla crop population. According to the authors, there were SSR markers previously developed for P. frutescens, but they further developed EST-SSR markers based on a transcriptome sequencing data. In the present study, two hundred of EST-SSR primer sets were designed to conduct PCR assays in two P. frutescens and two P. crispa lines, resulting in the amplifications of 84 stable PCR products among which 30 were polymorphic. Seventeen EST-SSR markers were applied for genotyping in the Perilla crop population to show its reliable population structure. An association test between these EST-SSR markers and visible seed phenotypes were performed in a simple way. Several comments for constructive criticism are left as follows. Hope that these would be useful to improve the scientific values of this study.
Major comments
(1) Novelty might be a critical issue of this manuscript. To clarify it, the authors have to describe what were previously done (which markers/how many markers were available) and what were scientifically delivered (how many new EST-SSR markers with their characteristics) in this study. Novelty may be able to further highlight by describing the availability of SSR markers in closely related species in the Labiatae family.
(2) How are the genic locations of the EST-SSR markers? Coding regions? UTR regions? Due to the structural restrictions, UTR regions would be more frequently polymorphic. It could be useful to add this information in the Result subsection 2.1 to obtain biological interpretation.
(3) There are a plenty of di-nucleotide EST-SSR sites. How the assayed 200 EST-SSR sites were selected? Any relevance to functional annotations to transcriptome contigs or anything else? In addition, how the 17 of EST-SSR makers were selected from the 30 polymorphic ones? In the text (in Line 114-115), there is one potential explanation was given while a clear selection criterion would be given with a logical story.
(4) The PCR assays for the 200 EST-SSR marker primer sets resulted in the amplification of 54 monomorphic bands and 30 polymorphic bands. How were the relationships between the cDNA length between a pair of primers and the PCR product length? It should be that the cDNA (transcriptome contig) length was identical or shorter than the corresponding PCR product. Then, the amplified sequences were verified by sequencing or not? It would be great if readers could see sequence alignments between the transcriptome contig sequences and the PCR product for the evaluation of the EST-SSR marker development procedures.
(5) The 17 EST-SSR markers were applied for genotyping of the CWTPC population. Based on the result, any EST-SSR markers were found to differentiate the two species (P. frutescens and P. crispa) or not? If yes, it might be useful for species differentiation of Perilla crop.
(6) The Results subsection 2.2 described the computational results related to the population structure and phylogenetic relationship of the South Korean CWTPC. What would be the biological message of this subsection? Is it the usefulness of the new EST-SSR markers or the population structure information of CWTPC? Since the interpretation was missing on this subsection, it was hard to understand when the reviewer read this part. Instead of it, it was written in Discussion subsection 3.2 (in Line 358-377). The reviewer feels that the interpretation (it does not include discussion contents) had better be written in Results section so that the readability would be quite improved.
(7) The association test between the EST-SSR markers and the visible seed characteristics sounds interesting and a good trial but it’s in a starting point. The number of the associated EST-SSR markers is many, so the reliability of the associations is questionable. In case the genotype-phenotype associations are valid, further data analysis would clarify the reliability; (1) Creation of genetic map and mapping the phenotype association, (2) principal component analysis (PCA) among the genotyping data, phenotyping data, and the taxonomic information together to represent a PCA biplot. If both of the data analyses are not useful, serious explanation about the association test results are needed to show a valid interpretation.
(8) Discussion section is too wordy and contains several general contents that would not be so useful in Original research article. One example is Line 234-243, it is like a paragraph in Introduction in a Review article of plant SSR markers and does not contain the main things of this study: Perilla crop. The contents in Discussion should include (1) novel EST-SSR markers for Perilla crop, (2) differentiation ability of the novel EST-SSR markers, (3) transferability/potential application of the novel EST-SSR markers, and (4) the diversity of CWTPC.
Regarding to transferability of SSR markers across species/genera/family and the application were reported in the following literatures.
Manoj K Rai et al. (2013) 40: 5067-71. doi: 10.1007/s11033-013-2608-1
Endo et al. Euphytica (2017) 213:56. doi: 10.1007/s10681-017-1846-z
The transferability of SSR markers is useful for species identification. A paper (Tuler et al. 2015) described an example in plant species.
Tuler et al. Mol Biol Rep (2015) 42: 1501-13. doi: 10.1007/s11033-015-3927-1
Minor comments
(1) The title looks fine? What is the most important content in this study? I wonder if that is complementing EST-SSR markers. Or, molecular diversity of CWTPC is important?
(2) The abstract lacks the result of association test for the seed phenotypes.
(3) In the caption of Table 2, the use of “Genetic diversity” is correct? PIC is an index of polymorphic status of DNA markers.
(4) Fig. 1 and Fig. 2 could be represented together because these are highly associated.
(5) Table 3 occupies a big space (too long). It could be changed into a wider format if this will be shown in the main text but not in Supplemental material.
(6) The text appearance in Line 283-316 looks different from other parts.
(7) In line 317 and 375, “Except for exceptional” and “exceptions of some exceptional”; except/exception/exceptional are redundantly used.
(8) Reference format is all correct? DOI URLs sometimes include underlines.
Author Response
Reviewer #3:
This manuscript reports new EST-SSR markers in Perilla frutescens and genotyping in a South Korean Perilla crop population. According to the authors, there were SSR markers previously developed for P. frutescens, but they further developed EST-SSR markers based on a transcriptome sequencing data. In the present study, two hundred of EST-SSR primer sets were designed to conduct PCR assays in two P. frutescens and two P. crispa lines, resulting in the amplifications of 84 stable PCR products among which 30 were polymorphic. Seventeen EST-SSR markers were applied for genotyping in the Perilla crop population to show its reliable population structure. An association test between these EST-SSR markers and visible seed phenotypes were performed in a simple way. Several comments for constructive criticism are left as follows. Hope that these would be useful to improve the scientific values of this study.
Major comments
(1) Novelty might be a critical issue of this manuscript. To clarify it, the authors have to describe what were previously done (which markers/how many markers were available) and what were scientifically delivered (how many new EST-SSR markers with their characteristics) in this study. Novelty may be able to further highlight by describing the availability of SSR markers in closely related species in the Labiatae family.
-> In the introduction, we described the development status of SSR primers in Perilla species as follows: In previous studies, developed Perilla SSR primers [19,21,24–26] have been used successfully to analyze the genetic variation and phylogenetic relationships as well as population structure between accessions of the CWTPC [5,7,11,18,19,21,22,27–29]. However, it is difficult to classify accessions of the CWTPC because the number of Perilla SSR primer sets (PSPSs) developed for analyzing the genetic variation and phylogenetic relationships of the CWTPC is still insufficient. See L. 76-82, in new version manuscript.
In addition, so far, the number of SSR primers developed from Perilla species is less than 200, and were almost developed by Korean researchers, mainly in our research team [Kim et al. 2021; Sa et al. 2018, 2019; Kwon et al. 2005; Park et al. 2008]. Therefore, more research on the development of SSR primers is needed for molecular genetics and breeding studies in Perilla species.
This study also selected and used the most germplasm accessions for the genetic diversity study of cultivated and weedy types of Perilla species in South Korea using a novel SSR markers. This has already been mentioned in the discussion and conclusion section as follow: The number of these accessions of the CWTPC is the most used for diversity analysis research in South Korea so far. Therefore, the results of this study obtained using the new PSPSs are expected to provide a useful information for understanding the genetic diversity and phylogenetic relationship of the accessions of the CWTPC in South Korea. See L. 280-284, 480-484, in new version manuscript.
(2) How are the genic locations of the EST-SSR markers? Coding regions? UTR regions? Due to the structural restrictions, UTR regions would be more frequently polymorphic. It could be useful to add this information in the Result subsection 2.1 to obtain biological interpretation.
-> In our previous study (Gene, doi:10.1016/j.gene.2015.01.028), we performed the de novo assembly of the PF98095 (cultivated var. frutescens) RNA-Seq data. We obtained 48,009 contigs with average length of 873 bp. A total of the 48,009 contigs were annotated using BlastX which is based on sequence similarity with known proteins of the nonredundant (NR) database in NCBI, 43,355 contigs (90.3% of the total contigs assembled) mapped possible homologous proteins from the NR database. Therefore, we intend to respond to your comments as our previous research paper (Gene, doi:10.1016/j.gene.2015.01.028).
(3) There are a plenty of di-nucleotide EST-SSR sites. How the assayed 200 EST-SSR sites were selected? Any relevance to functional annotations to transcriptome contigs or anything else? In addition, how the 17 of EST-SSR makers were selected from the 30 polymorphic ones? In the text (in Line 114-115), there is one potential explanation was given while a clear selection criterion would be given with a logical story.
-> In our previous studies (Gene, doi:10.1016/j.gene.2015.01.028; Genes. Genom. doi:10.1007/s13258-018-0727-8), we searched microsatellite sites in the assembled transcriptome sequences of PF98095 (cultivated var. frutescens). The SSRs with di-, tri-, and tetra-nucleotide repeat units were identified, and designed the primer pairs based on the SSR flanking sequences. We searched all unigenes in the PF98095 and detected 15,991 SSR loci. We tested di-nucleotide SSRs first due to more variation than tri- or tetra-nucleotide SSRs.
In addition, in our study, among the 30 SSR primers selected in the preliminary experiment, we selected 17 SSR primers for genetic diversity analysis among germplasm accessions of the CWTPC, because they showed the most distinct amplification patterns and polymorphisms between the four accessions of the CWTPC, and finally used them for genetic diversity analysis for 90 accessions of the CWTPC collected from South Korea. Therefore, in the revised manuscript, we explained ‘why we chose 17 SSR marker sets from 30 SSR marker sets’. In addition, we presented Supplementary Figure 1, showing the amplification pattern of the 200 SSR primers developed in this experiment. See L. 107-111, in new version manuscript.
(4) The PCR assays for the 200 EST-SSR marker primer sets resulted in the amplification of 54 monomorphic bands and 30 polymorphic bands. How were the relationships between the cDNA length between a pair of primers and the PCR product length? It should be that the cDNA (transcriptome contig) length was identical or shorter than the corresponding PCR product. Then, the amplified sequences were verified by sequencing or not? It would be great if readers could see sequence alignments between the transcriptome contig sequences and the PCR product for the evaluation of the EST-SSR marker development procedures.
-> In our study, the results of this study are presented as Supplement Tables and Figures. The development of the SSR primers was carried out in the same way as our research team published in previous study. Among them, Supplement Fig. 1 showed that an example of an SSR profile of four accessions of cultivated and weedy types Perilla crop (the order is cultivated var. frutescens, weedy var. frutescens, cultivated var. crispa, weedy var. crispa) using 200 newly designed SSR primer sets. The two pictures are examples of SSR profiles for 120 of 200 primer sets. Supplement Fig. 2 showed that an example of an SSR profile of 90 Perilla accessions collected from South Korea run on 6% acrylamide nature gel, using the SSR primers KNUPF133 (a) and KNUPF141 (b). Supplement Table 1 showed that low data of sequence information for each Perilla SSR primer set of the 30 developed Perilla SSR primers. In particular, in Table 1, we presented in detail the nucleotide sequence information and lengths of 30 SSR primers developed in Perilla species. See Supplement Table 1 and Supplement Figures 1 and 2.
(5) The 17 EST-SSR markers were applied for genotyping of the CWTPC population. Based on the result, any EST-SSR markers were found to differentiate the two species (P. frutescens and P. crispa) or not? If yes, it might be useful for species differentiation of Perilla crop.
-> Among the primers used for analysis, there were no markers that clearly discriminated between cultivated and weedy types. However, analyzes using all allelic fragments amplified with these primers were able to distinguish between cultivated and weedy types. However, some exceptional lineages are not clearly differentiated between cultivated and weedy types. As mentioned in the discussion section, these results appeared to have a gene flow or gene exchange due to natural crossing between cultivated and weedy types of Perilla crop. See L. 383-385, in new version manuscript.
(6) The Results subsection 2.2 described the computational results related to the population structure and phylogenetic relationship of the South Korean CWTPC. What would be the biological message of this subsection? Is it the usefulness of the new EST-SSR markers or the population structure information of CWTPC? Since the interpretation was missing on this subsection, it was hard to understand when the reviewer read this part. Instead of it, it was written in Discussion subsection 3.2 (in Line 358-377). The reviewer feels that the interpretation (it does not include discussion contents) had better be written in Results section so that the readability would be quite improved.
-> Based on the reviewer’s comments, we described the reasons for studying the population structure and phylogenetic relationship of the South Korean CWTPC as follow: To further understand the phylogenetic relationships of CWTPCs collected from Korea, we analyzed population structure and phylogenetic relationships among accessions of the CWTPC using newly developed PSPSs. See L. 161-163, in new version manuscript.
(7) The association test between the EST-SSR markers and the visible seed characteristics sounds interesting and a good trial but it’s in a starting point. The number of the associated EST-SSR markers is many, so the reliability of the associations is questionable. In case the genotype-phenotype associations are valid, further data analysis would clarify the reliability; (1) Creation of genetic map and mapping the phenotype association, (2) principal component analysis (PCA) among the genotyping data, phenotyping data, and the taxonomic information together to represent a PCA biplot. If both of the data analyses are not useful, serious explanation about the association test results are needed to show a valid interpretation.
-> Thank you for your comments. However, among the two methods presented by you, the method of Creation of genetic map and mapping the phenotype association is impossible. Because the chromosomal location information of SSR primers developed in Perilla crop is not known yet. In addition, the results of the Principal Component Analysis (PCA) analysis are almost identical to those of the NTSYS analysis. Therefore, we cannot prove whether both of the data analyses are not useful.
Instead, we added for the association test results in the Discussion section as follow: Although, these SSR markers are thought to be useful molecular markers for distinguishing seed characteristics in the CWTPC, precise analysis using genetic mapping population or more research materials is required in the future. See L. 394-397, in new version manuscript.
If you think that this explanation is insufficient, we will delete the 2.3 Seed characteristics and association mapping analysis of 90 accessions of the CWTPC using novel PSPSs.
(8) Discussion section is too wordy and contains several general contents that would not be so useful in Original research article. One example is Line 234-243, it is like a paragraph in Introduction in a Review article of plant SSR markers and does not contain the main things of this study: Perilla crop. The contents in Discussion should include (1) novel EST-SSR markers for Perilla crop, (2) differentiation ability of the novel EST-SSR markers, (3) transferability/potential application of the novel EST-SSR markers, and (4) the diversity of CWTPC.
- Manoj K Rai et al. (2013) 40: 5067-71. doi: 10.1007/s11033-013-2608-1
- Endo et al. Euphytica (2017) 213:56. doi: 10.1007/s10681-017-1846-z
The transferability of SSR markers is useful for species identification. A paper (Tuler et al. 2015) described an example in plant species.
- Tuler et al. Mol Biol Rep (2015) 42: 1501-13. doi: 10.1007/s11033-015-3927-1
-> Based on the reviewer’s comments, we deleted Line 234-243 in old manuscript. And also, in the discussion section, we deleted some sentences that is not directly related to this study. See L. 235-254, 297-325, in new version manuscript.
Minor comments
(1) The title looks fine? What is the most important content in this study? I wonder if that is complementing EST-SSR markers. Or, molecular diversity of CWTPC is important?
-> Based on the reviewer’s comments, we changed the current title ‘Genetic diversity and phylogenetic relationships among accessions of cultivated and weedy types of Perilla frutescens (L.) Britton collected from South Korea using newly developed SSR markers’ to the following title ‘Utilization of novel Perilla SSR markers to evaluate genetic diversity of native Perilla germplasm accessions collected from South Korea’. See L. 2-4, in new version manuscript.
(2) The abstract lacks the result of association test for the seed phenotypes.
-> We added the result of association test for the seed traits in the abstract. See L. 25-27, in new version manuscript.
(3) In the caption of Table 2, the use of “Genetic diversity” is correct? PIC is an index of polymorphic status of DNA markers.
-> Yes, in the caption of Table 2, the use of “Genetic diversity” is correct.
(4) Fig. 1 and Fig. 2 could be represented together because these are highly associated.
-> Based on the reviewer’s comments, Fig. 1 and Fig. 2 combined and changed to Fig. 1. And also Fig. 2 changed to Fig. 1. See Fig. 1 and Fig. 2, in new version manuscript.
(5) Table 3 occupies a big space (too long). It could be changed into a wider format if this will be shown in the main text but not in Supplemental material.
-> We modified Table 3 to not take up too much space (too long). See Table 3, in new version manuscript.
(6) The text appearance in Line 283-316 looks different from other parts.
->We checked the text in Line 283-316 in old version manuscript. We didn't find any problems. See L. 276-296, in new version manuscript.
(7) In line 317 and 375, “Except for exceptional” and “exceptions of some exceptional”; except/exception/exceptional are redundantly used.
-> We changed “Except for exceptional” and “exceptions of some exceptional” to “Except for ambiguous” and “exceptions of some ambiguous”, respectively. See L. 342, 347, in new version manuscript.
(8) Reference format is all correct? DOI URLs sometimes include underlines.
->We modified the Reference format correctly. See L. 529-641, in new version manuscript.
Thank you for your review comments.

Round 2
Reviewer 3 Report
The authors presented a study on the development of new SSR markers in Perilla species with the applications for assessing the genetic diversity and association mapping in a South Korean Perilla crop population. According to the authors argument, previously developed SSR markers are insufficient to analyze Perilla germplasms in South Korea. The outcomes of this study can promote understanding of the genetic diversity and genetic analyses in not only for South Korean Perilla germplasms but also for other Perilla materials in the future.
The revised manuscript is now better in shape. However, English had better be improved for reader-friendliness before publication. I would like to suggest the authors to review the terminology of several scientific words/expressions to document their scientific outcomes properly. I’ve left several comments associated with the English issues below as examples.
Throughout the manuscript, Perilla SSR primer sets (PSPSs) were used. I agree with the use of this in case of Line 14, but in Line 16, there is a phase “30 PSPSs were newly developed”. It exactly means that 30 Perilla SSR markers were developed. I understand that SSR primers were designed based on a transcriptome assembly, but transcriptome contigs sometimes contain mis-assemblies and sequence errors due to the technical limitation. Hence, “EST-SSR marker development” often needs experimental verification by genomic PCR with sequencing of amplified PCR products. In the revised manuscript, verification was not presented by sequencing of PCR products. Hence, it’s hard to mention clearly that this study identified SSR/SSR polymorphisms as in the heading in Line 101. In case there is evidence for SSR size polymorphisms for at least a few PSPSs, the present expression sounds proper. Furthermore, primers designed from transcript sequences might not fit to the corresponding genomic sequences because transcripts lack introns. So PCR amplification test in genomic DNA is meaningful to verify the appropriateness of primer designs from transcript sequences. Even though the 54 PSPSs showed monomorphic bands, these may be polymorphic when other Perilla materials were used.
In Line 16 and Line 83, the two technical words “transcriptome sequencing” and “RNA-sequencing” are used in the same sentence, respectively. Here it has to be noted that these two words have almost the same meaning. A suggested modifications for the two parts are (1) in Line 16, 30 PSPSs were newly developed based on transcriptome contigs in P. frutescens, and (2) in Line 83, Recently, RNA sequencing has presented…
In Line 18-19, what is the intended meaning of “from all loci”?
In Line 25, By association analysis of the 17 PSPSs….. three seed traits in the 90 Perilla accessions
In Line 76-82, this part seems unclear a bit. In other words, the application of the previously available SSR markers was not insufficient to identify Perilla germplasms in South Korea? Or the number of SSR markers is quite less (how many available exactly?), hence more SSR markers are needed for analyses of genetic diversity of Perilla? Moreover, by using the new SSR markers, classification of the Perilla germplasm were successfully and sufficiently made or not? These (novelty and significance) had better be clarified seriously.
The first paragraph of 2.1 (particularly in Line 107-116) is hard to read smoothly. “Among the 30 PSPSs, 17 PSPSs presented clear band patterns that differentiate the Perilla species.” That’s it if it is right.
In Line 153, the term “Genetic diversity” is used in the Table title. According to the manuscript context, it was referred based on three molecular diversity indices (MAF, GD, and PIC). Here it is questioned what this result (called as genetic diversity) means. I understand that Table 2 with Line 139-152 represents the three diversity indices among the three subpopulations. Then what is the message brought from these results?
Reference format is all correct? It looks there are multiple fonts were used. Non-abbreviated journal name “PLoS ONE” contains a dot after “PLoS” in Line 596. Some references are with DOI, but some are without DOI.
Line 512 Table 1: Raw data of …
Author Response
The revised manuscript is now better in shape. However, English had better be improved for reader-friendliness before publication. I would like to suggest the authors to review the terminology of several scientific words/expressions to document their scientific outcomes properly. I’ve left several comments associated with the English issues below as examples.
Throughout the manuscript, Perilla SSR primer sets (PSPSs) were used. I agree with the use of this in case of Line 14, but in Line 16, there is a phase “30 PSPSs were newly developed”. It exactly means that 30 Perilla SSR markers were developed. I understand that SSR primers were designed based on a transcriptome assembly, but transcriptome contigs sometimes contain mis-assemblies and sequence errors due to the technical limitation. Hence, “EST-SSR marker development” often needs experimental verification by genomic PCR with sequencing of amplified PCR products. In the revised manuscript, verification was not presented by sequencing of PCR products. Hence, it’s hard to mention clearly that this study identified SSR/SSR polymorphisms as in the heading in Line 101. In case there is evidence for SSR size polymorphisms for at least a few PSPSs, the present expression sounds proper. Furthermore, primers designed from transcript sequences might not fit to the corresponding genomic sequences because transcripts lack introns. So PCR amplification test in genomic DNA is meaningful to verify the appropriateness of primer designs from transcript sequences. Even though the 54 PSPSs showed monomorphic bands, these may be polymorphic when other Perilla materials were used.
-> Thank you very much for your comments on our manuscript. In the revised manuscript, we changed SSR identification and polymorphism to Polymorphic test for new developed PSPSs in the 2.1. heading title. See L 101, in new version manuscript.
Unfortunately, we did not verified sequencing for amplified alleles, but 30 newly developed PSPSs showed good amplication patterns and polymorphisms in different types (two cultivated types and their weedy types of Perilla crop) of P. frutescens. See Supplement Figure 1. Therefore, we confirmed the appropriateness of designed primers from transcriptome contigs. We will try to verify sequence for alleles from new designed primers in further research.
In Line 16 and Line 83, the two technical words “transcriptome sequencing” and “RNA-sequencing” are used in the same sentence, respectively. Here it has to be noted that these two words have almost the same meaning. A suggested modifications for the two parts are (1) in Line 16, 30 PSPSs were newly developed based on transcriptome contigs in P. frutescens, and (2) in Line 83, Recently, RNA sequencing has presented
-> We changed these sentences according to your suggestion. “30 PSPSs were newly developed based on the Perilla crop transcriptome contigs”. See L. 16, in new version manuscript. And also, “Recently, RNA-sequencing has presented……”. See L. 83, in new version manuscript.
In Line 18-19, what is the intended meaning of “from all loci”?
-> We changed “from all loci” to “selected 17 PSPSs”. See L. 19, in new version manuscript.
In Line 25, By association analysis of the 17 PSPSs….. three seed traits in the 90 Perilla accessions
-> We changed to “Based on association analysis between 17 PSPSs and three seed traits in 90 Perilla accessions”. See L. 25-26, in new version manuscript.
In Line 76-82, this part seems unclear a bit. In other words, the application of the previously available SSR markers was not insufficient to identify Perilla germplasms in South Korea? Or the number of SSR markers is quite less (how many available exactly?), hence more SSR markers are needed for analyses of genetic diversity of Perilla? Moreover, by using the new SSR markers, classification of the Perilla germplasm were successfully and sufficiently made or not? These (novelty and significance) had better be clarified seriously.
-> The number of SSR primers developed from Perilla species is less than 200 in previous studies, and were almost developed by Korean researchers, mainly in our research team [Kim et al. 2021; Sa et al. 2018, 2019; Kwon et al. 2005; Park et al. 2008]. Therefore, the evaluation of the genetic variability of the collected Perilla resources was very limited. It was very difficult to distinguish between cultivated and weed type of Perilla or var. frutescens and var. crispa using previously developed markers. For this reason, our research team wanted to develop more molecular marker for further researches, such as construction of genetic and physical map and whole genome sequencing.
The first paragraph of 2.1 (particularly in Line 107-116) is hard to read smoothly. “Among the 30 PSPSs, 17 PSPSs presented clear band patterns that differentiate the Perilla species.” That’s it if it is right.
-> Based on your comments, we changed the first paragraph of 2.1 (particularly in Line 107-116) for readability as follow: “Among the 30 PSPSs, 17 PSPSs presented clear band patterns that differentiate the Perilla species.”. See P. 3, L. 108-112, in new version manuscript.
In Line 153, the term “Genetic diversity” is used in the Table title. According to the manuscript context, it was referred based on three molecular diversity indices (MAF, GD, and PIC). Here it is questioned what this result (called as genetic diversity) means. I understand that Table 2 with Line 139-152 represents the three diversity indices among the three subpopulations. Then what is the message brought from these results?
-> We changed “Genetic diversity” to “Genetic variation” in the Table 2 title. See L. 155, in new version manuscript. In addition, information on the genetic variation of genetic resource collections is very important for both the conservation and utilization of crop germplasms. There are weed types with various genetic variation in Perilla crop. Therefore, genetic variation (or diversity) analysis of the collected resources must be preceded to find a unique allele in a breeding program. So, we confirm genetic diversity for different three types of Perilla crop. We find out that weedy type of var. frutescens had higher genetic diversity than other types based on three diversity indices (MAF, GD, and PIC) in South Korea.
Reference format is all correct? It looks there are multiple fonts were used. Non-abbreviated journal name “PLoS ONE” contains a dot after “PLoS” in Line 596. Some references are with DOI, but some are without DOI.
-> We changed that all References is correct. See L. 514, in new version manuscript.
Line 512 Table 1: Raw data of …
-> We deleted the phrase ‘Low data of’ from the title of Supplement Table 1
Thank you for your review comments.
